# Do Strategic Human Resources and Artificial Intelligence Help to Make Organisations More Sustainable? Evidence from Non-Governmental Organisations

**Amal Alnamrouti [1], Husam Rjoub [2,3,]*\* and Hale Ozgit [4]**

[1] Department of Business Administration, Faculty of Economics and Administrative Sciences, Cyprus International University, Mersin 10, Haspolat 99040, Turkey; amalalnamrouti@gmail.com
[2] Department of Accounting and Finance, Faculty of Economics and Administrative Sciences, Cyprus International University, Mersin 10, Haspolat 99040, Turkey
[3] Business Administration Department, Faculty of Management Sciences, ILMA University, Karachi 75190, Pakistan
[4] Department of Tourism and Hospitality Management, Cyprus International University, North Cyprus, Mersin 10, Nicosia 99258, Turkey; hozgit@ciu.edu.tr
\* Correspondence: hrjoub@ciu.edu.tr

**Abstract:** Uncertainty and a lack of stability are among the difficulties non-governmental organisations face. However, certain strategies for ensuring their performance's sustainability have not been empirically demonstrated in the literature. Using strategic resource management practises and artificial intelligence, this study examines the effect of organisational learning and corporate social responsibility on the sustainability of non-governmental organisations' performance. The survey gathered data from 171 participants representing 21 United Nations organisations and 70 non-governmental organisations in Jordan to accomplish this goal. The data were analysed using WarpPLS and PLS-SEM. The study demonstrates that organisational learning, artificial intelligence, strategic human resource management practises, and corporate social responsibility all contribute to the long-term viability of non-governmental organisations. Furthermore, the study discovered that strategic resource management practises and artificial intelligence significantly mediate the relationship between organisational learning and sustainable organisational performance on the one hand, and corporate social responsibility on the other. Finally, the study provides theoretical and practical guidance on how to apply the findings to assist non-profit organisations' management in utilising organisational learning, corporate social responsibility, artificial intelligence, and strategic resource management practices to help them run their internal operations in a more efficient and sustainable manner over time.

**Keywords:** artificial intelligence; corporate social responsibility; sustainability; non-governmental organisations; sustainable performance; PLS-SEM; Jordan

## 1. Introduction

According to [1] the dynamic and competitive nature of today's business environment has elevated organisational learning (OL) to a core capability of high-performing organisations and a primary component of corporate strategy. The literature indicates that promoting OL has a significant impact on upgrading and transforming the national economy, as OL demonstrates efforts to develop knowledge assets and proposes practical methods for managing them [2]. Additionally, Refs. [3,4] suggested that learning is the "next competitive advantage source" or "only competitive advantage source" [4,5], and, as [6] suggested, "the cornerstone to a company's future success", making it imperative and critical for both scholars. Meanwhile, society has expressed concern about the devastation caused by commercial activities [7], and some businesses have expressed concern about the extent to which all actors are treated ethically and responsibly, as well as the

impact of the "stakeholder theory" on organisational performance [6]. Despite the obvious financial costs associated with balancing the economic interests of stakeholders with the ethical, environmental, and social concerns of other stakeholders, some leading corporate executives are increasing their corporate social responsibility (CSR) investments [8,9]).

Corporate executives believed that the benefits of CSR practises such as increased employee morale, increased customer loyalty, and other forms of social capital outweighed the costs of socially responsible actions, according to [10]. Nonetheless, several researchers assert that a firm's performance in today's complex, irregular, and discontinuous business environment has necessitated the search for a strategy to ensure a "competitive edge" that encompasses not only product, innovation, or resources, but also the ability to generate novel and useful knowledge [9,11]. Additionally, Ref. [12] stated that "to survive and grow, modern businesses must learn faster and more efficiently than competitors." According to [13], organisational learning (OL) is "a process that lays the groundwork for the development of a knowledge-based economy". Meanwhile, studies indicate that "intellectual capital and knowledge" are required in the modern world, rather than "physical capital" [14,15], as are "information technologies" (ITs), which have permeated professional activities, causing disruption and influencing all critical operations and procedures [16]. Ref. [17] asserts that by integrating business processes and information technology, a visible effect on the business ecosystem, particularly the nexus between firms and their prospects, customers, and partners, can be achieved. Additionally, information technology is critical for advancing a business's processes and operations. According to [18,19], artificial intelligence (AI) is still the most important application of information technology in the modern world due to its rapid growth.

As revealed by the literature review (see Table 1), numerous previous studies attempted to determine the effect of OL on organisational performance [20–23]. Meanwhile, the findings have been inconsistent, emphasising the importance of identifying the mediating variables to ensure more effective OL promotion. This position is consistent with [24] conclusion that the mechanism by which "organisational learning" contributes to "organisational performance" (OP) requires additional theory. Additionally, internal procedures have been implemented to improve the performance of "firm-level resources" and "capacity". In contrast to [25], the employee is viewed as a critical source of information for achieving sustainable organisational performance (SOP), which can be accomplished only through "strategic human resource management" (SHRM). Therefore, according to this research, SHRM should be included in the study because it may collaborate with OL and CSR to achieve SOP, making it a possible moderator of the effect of OL and CSR on SOP.

According to [25], "SHRM is a management strategy that contributes to an organization's performance improvement through the systematic integration of several best human resource practises". Meanwhile, while successful implementation of OL and CSR requires the support of an internal managerial approach, the importance of this approach has not been fully recognised [4,25]. The majority of research has concentrated on the factors that contribute to the success of for-profit businesses, with less attention paid to non-governmental organisations (NGOs).

On the other hand, Refs. [12,26] define NGOs as autonomous organisations or institutions whose primary mission is to volunteer in civil society in order to counteract the state's failures. Non-governmental organisations face a number of obstacles to success, including a lack of stability and uncertainty [27]. NGOs must effectively manage their resources [28]. These are the organisational structures that enable two distinct goals to be pursued: individual (self-realisation) and social (satisfaction and responsibility) [29]. This approach places a premium on sustainability because it necessitates a delicate balance of resources: funding, new partnerships, new revenue streams, professional human resources, and evaluation and control systems [30]. Non-governmental organisations experiment in novel ways [31]. The relationship between organisational innovation and sustainability is critical for ensuring sustainable development. Concerns about sustainability serve as a catalyst for novel, innovative approaches [32]. Thus, the literature indicates that achieving

organisational goals for sustainable development requires innovation [32], which is also true for NGOs. Given the preceding, the increased emphasis on organisational performance by various stakeholders in order to maintain a sustainable competitive edge in the markets in which they operate, and the fact that various determinants of organisational performance have been investigated in the literature, the purpose of this research is to examine OL and CSR as drivers of standard operating procedures (SOP) in non-governmental organisations (NGOs) through the lens of strategic human resource management (SHRM) and artificial intelligence (AI), both of which have received little attention in the literature. Thus, the study contributes to the body of knowledge by examining the role of organisational learning, artificial intelligence, SHRM practises, and CRS in promoting sustainable organisational performance in non-governmental organisations. Additionally, the study's findings have theoretical and practical implications for non-governmental organisation (NGO) management in terms of enhancing sustainability performance.

The following sections comprise the remainder of the paper: the next section discusses how the study's major variables were conceptualised and the pertinent prior research that resulted in the development of study hypotheses. Next, the methodology section delves into the data collection and source and the analysis method. Finally, the data analysis and the conclusion discuss the findings and implications.

**Table 1.** Prior research findings related to the constructs observed in this study.

| Authors | Variables Tested | Findings |
|---------|------------------|----------|
| [20] | Organisational learning, business guanxi, and strategic performance | The findings concluded that when operating business in an emerging economy such as China, a right fit between organisational learning and guanxi networking can result in higher level of strategic performance. |
| [21] | Organisational learning capacity, entrepreneurial orientation, and small business performance | The study reported that a significant positive relationship between organisational learning capacity and entrepreneurship orientation. It was also reported that a positive relationship between entrepreneurial orientation and sales and market share growth. |
| [25] | Strategic human resource management practices, human capital development, employee commitment, and sustainable competitive advantage | The study reported that strategic HRM has a positive effect on sustainable competitive advantage. The study also concluded that employees' commitment partially mediates the relationship between strategic HRM practices and sustainable competitive advantage. |
| [32] | Organisational learning capacity, knowledge sharing, human resource cost, adoption of environmental practices, and product innovation performance | The study reported that social and environmental developments are two crucial antecedents of product innovation performance. It was also reported that they both contribute to different pathway that result to product innovation performance. |
| [33] | Artificial intelligence and business process management | The study reported that deploying AI within an organisation would aid the organisation in automating inquiries and advice related to quality management, supply chain management, and fleet asset management. |
| [34] | Corporate social responsibility and organisational performance | The study reported a positive effect of CSR on the non-financial and financial performance in the context of the Jordanian telecommunication firms. |

## 2. Literature Review and Hypotheses Development

### 2.1. Organisational Learning (OL)

In recent years, the OL concept has garnered considerable attention from researchers and practitioners seeking to improve their organisations [35]. Its application in theory is regarded as a dynamic concept, emphasising the continuous changes that characterise firms [35]. The term "organisational learning" was coined in 1978 by [36], who initially defined it as "the practise of enhancing actions by gaining a better grasp of the situation". Other authors later defined it as "the practise of enhancing actions by gaining a better grasp of the situation" [37]. Ref. [38] define OL as "the capacity of an organization's capability or procedures to sustain or increase performance based on past performance", while another study defines it as "the processes by which firms develop, supplement, and organise the knowledge and procedures that surround their activities, as well as adapting or developing firm efficiency through improved utilisation of their workforce's general skills" [39]. Additionally, Ref. [40] defined OL as "an organization's capacity to process knowledge—that is, to create, absorb, transmit, and integrate knowledge—as well as to adapt its behaviour to the current cognitive environment in order to improve performance". In contrast to Sanzo et al. (2012), this article views OL as a highly active process in which individuals create, consume, and integrate knowledge based on information obtained from external sources in order to improve the firm's resources and capabilities.

According to [23,41], OL is more than the sum of individual knowledge. According [41], the organisation creates a cohesive system and establishes organisational routines by allowing members to acquire, interpret, and distribute information. As a result, OL has been recognised as a critical component of enabling organisations to achieve competitive advantages and improve performance [5]. Additionally, Ref. [42] asserted that OL can help an organisation broaden its knowledge base and strengthen the talents and skills that promote creative thinking and behaviour.

### 2.2. Corporate Social Responsibility (CSR)

CSR is regarded as a significant concept in the literature from theoretical and practical perspectives, prompting several attempts by scholars to define it [43]. Numerous arguments have been advanced, and certain factors have been agreed upon as defining CSR. Refs. [44,45] defined CSR as a business practise or activity that ensures an organisation's compliance with multiple obligations to various stakeholders, including employees, shareholders, customers, the environment, and local communities regarding the firm's business activities and procedures. Refs. [44,46] asserts that this obligation encompasses both the philanthropic and ethical, economic, and legal dimensions of the social expectations that an organisation must meet. There is no doubt that CSR as a concept is a non-starter among academics and practitioners, as a consensus definition appears to be lacking in the literature. Ref. [47] note that the term "corporate social responsibility" is used differently in different organisations, including "corporate social responsiveness", "ethical business practises", "corporate citizenship", "corporate sustainable business practises", and "stakeholder management". According to [48], CSR is "business's commitment to sustainable economic development through collaboration with employees, their families, the local community, and society to improve people's quality of life".

Additionally, CRS was defined as a concept that enables businesses to incorporate social and environmental concerns into their operations and interactions with other stakeholders [49]. Similar to this definition, the "World Business Council for Sustainable Development" (CSR) defines CSR as a business's ongoing emphasis on ethical conduct in order to advance economic growth while improving the living standards of employees, dependents, the local community, and society as a whole [50]. Similarly, according to [51], CSR is defined as "a comprehensive spectrum of fundamentals that firms are expected to recognise and embody in their operations". This includes "regard for life rights, suppliers, fair worker treatment, consumers, being a responsible corporate citizen in the communities in which they operate, and environmental protection".

From a global perspective, Ref. [52] definition, which implies a multidimensional approach to CSR practises, suggests that CSR is related to the expectations of society's constituent groups regarding the organisational behaviour that the firm must identify and attempt to conform to. Furthermore, according to [53], successful company management now requires high attention to several aspects of business performance and strategic engagement of internal and external stakeholders. In keeping with this perspective, this research considers the internal stakeholder, as we believe that CSR activities focused on employees have the greatest potential to significantly impact the firm. Internally, CSR is concerned with health and safety, social capital investment, and fair and empowered human resource management [53].

### 2.3. Sustainable Organisational Performance (SOP)

It is impossible to overstate an organisation's contribution to a nation's wealth. As a result, successful managers constantly look for ways to grow, enhance, and sustain their business, particularly in a developing economy. Successful managers have mastered the art of problem solving in order for their organisations to overcome and survive obstacles and progress toward long-term viability, profitability, and progress. As a result, Refs. [54,55] observed an increasing interest among academics and practitioners in examining organisational performance aspects, taking into account the critical determinant factors [7,15,39,56,57]. Meanwhile, some studies examine the effects of human resources, strategy, and operations on traditional organisational performance, with relatively few examining organisational performance [55,58]. According to [55], performance can be defined in several ways: "the ability to accomplish something with a specific goal in mind; the outcome of an activity; the capacity to accomplish or the possibility of accomplishing a goal". This position is consistent with [59] findings, who concluded that performance can be interpreted differently depending on the individual. This can be accomplished in various ways when assessing a business's performance. Ref. [60] defines performance as the sum of an individual's actions and accomplishments compared to competitors.

Based on this performance description, an organisation's performance can be evaluated using non-financial and financial factors [61]. According to the literature, Ref. [62] identified five major performance classification factors: market/customer, financial performance, human resource development, process, and years to come. Nonetheless, product market, shareholder, and financial results are the most frequently used metrics for assessing an organisation's performance and ability to meet its goals and objectives [54]. Despite this [63] identified three distinct outcomes for organisational performance: organisational outcomes such as quality, efficiency, and production; human resource outcomes such as satisfaction, behaviours, commitment, and attitudes; and financial outcomes such as profit and market share. Given these factors, Ref. [10] define sustainable organisational performance (SOP) as "the foundation's capacity to meet the needs of its stakeholders while also enhancing investment and management plans and strategies to ensure future earnings, a sustainable environment, and social welfare". As a result, Ref. [64] asserted that sustainable organisational performance occurs when management can develop strategies to increase market share, talent, stakeholder profit, and so on, while simultaneously lowering operational costs and employee turnover. The study concludes by stating that an organisation is deemed to be sustainable when it strives to maintain a low level of external danger and internal change [64].

### 2.4. Strategic Human Resource Management (SHRM)

In today's business world, human capital is regarded as a significant resource due to the employee's possession of tacit knowledge, which an organisation can use to achieve a competitive edge in the market in which they operate [65,66]. As proposed in the literature, the "valuable, rare, imitable, and non-substitutable" (VRIN) requirements of a competitive resource (Barney, 1991) were met by the described attribute of human capital, thereby establishing SHRM as a critical characteristic of today's competitive enterprises.

Ref. [67] define SHRM as "a management attitude that ensures human resources are used to add value to the business by providing a competitive edge, thereby achieving the organization's goals, purpose, and vision". SHRM as a concept dates back to the 1990s, with an emphasis on an integrative, value-driven strategy and proactive approach to human resource management (HRM) that emphasises issues such as aligning HRM practises with firm strategic goals and integrating HR processes into senior leadership [68]; performance evaluation and the value added to firm performance by HRM [67,68]. Additionally, SHRM practise is defined as "the structure of plans that the firm's human resources department implements to accomplish the organization's objectives" [69]. According to some studies, SHRM practises establish a link between business requirements and firm activity through their application approach [70,71]. Additionally, the practise unifies and guides employees who are not aligned with the organisation's business strategies [72], while also assisting the organisation in achieving a competitive advantage [73,74]. Similarly, the literature suggests that when SHRM practises are implemented in a business, they assist the company in gaining an unmatched competitive edge over competitors.

According to the description of SHRM practises, additional internal factors can affect a firm's performance. This position is consistent with the findings of [70], who asserted that SHRM should be viewed as a strategy implemented to motivate, enhance, and reduce employee turnover to ensure effective implementation and success of both the employee and the organisation. This position is consistent with [75] study, which demonstrates that human resource strategies significantly impact significant positive organisational outcomes. Additionally, the literature indicates that adopting and integrating strategy decisions into human resource management systems is a significant distinction between SHRM and HRM [55,76,77]. Based on a description of SHRM and SHRM practises, this study argues that SHRM is a critical and valuable asset to every organisation, which is believed to be scarce, unique, and occasionally difficult to replicate or substitute, and one that an organisation can leverage to achieve sustainable performance.

*2.5. Artificial Intelligence (AI)*

Without a doubt, AI is the most amazing application of "information technology" (IT) [15], a technology believed to have advanced tremendously over the last decades [15,18,19]. Ref. [15] define AI as "a collection of beliefs and strategies for developing robots that mimic human intelligence". By comparison, Ref. [78] discovered that it is a frequently used term to refer to the use of a computer to mimic intelligent behaviour with minimal human intervention. In summary, technological devices to replicate human cognitive abilities to achieve a firm objective autonomously, despite any obstacles encountered, are best described as artificial intelligence [15,78]). Refs. [79,80] The rapid growth of AI is due to significant advancements in computer computational capability and access to massive data sets [79,80]. These studies discovered that AI and its technologies significantly impacted how businesses and firms operate. To put it succinctly, the entire structure of artificial intelligence has altered the way businesses operate and interact with their environments. According to [19,33,81], AI takes a novel approach to information management, which represents both a challenge and a tremendous opportunity for businesses; however, realising this potential requires a shift in culture, mindset, and capabilities. According to these findings, Refs. [79,82] recommend deploying AI throughout an organisation's value chain, allowing for the integration of all elements such as research and development, preservation, marketing and sales, operations, production scheduling, demand and prediction, and services. With AI as a primary driver of growth, the literature highlights several notable accomplishments that the deployment of AI can enable an organisation to achieve. Refs. [79,82], for example, identified improvements in operational efficiency, maintenance and supply chain operations, customer experience optimisation and enhancement, product and service development, and the item recommendation process. Similarly, Refs. [79,82] believe that AI will enable faster and more automatic adaptation to changing market condi-

tions, develop new business models, and optimise the supply–demand nexus and more efficient forecasting and planning capacity.

Additionally, according to [79,80], deploying AI enables fraud detection, automates threat intelligence, information systems, and sales process optimisation. Refs. [83,84]), in addition to pharmacological vigilance, suggested the diagnosis and treatment of pathologies, the prediction of disease and its evolution, the promotion of individualised treatments, and assisting people in making decisions about diagnosis and prevention through epidemic anticipation. Finally, Refs. [33,81] concluded that deploying AI within an organisation would aid the organisation in automating inquiries and advice related to quality management, supply chain management, and fleet asset management.

### 2.6. Development of Hypotheses

### 2.6.1. Organisational Learning and Sustainable Organisational Performance

According to [12], organisational learning (OL) remains a significant source of competitive advantage in the context of strategic management, which is believed to be necessary for a firm to ensure the sustainability of its performance. According to [85], "the ability to learn faster than your competitors may be your only competitive advantage". "The rate at which individuals and groups acquire knowledge may become the sole source of sustained competitive advantage, especially in knowledge-intensive industries" [86]. Numerous interpretations of OL imply a theoretical connection between OL and firm performance [38,87–89]. According to these studies' definitions, OL directly and indirectly contributes to organisational performance (OP). The relationship between OL and OP has been investigated empirically in a few studies [35,90–95]. For example, in their study on the impact of OL innovation on firm financial performance, Ref. [90] discovered a positive, significant relationship between the two variables. The finding is consistent with [94], who established that cohesion of teamwork and OL improves OP in Spanish firms. This finding is consistent with the findings of [92], who established a positive and significant effect of OL on OP in Spanish firms, as well as [96]), who discovered a significant relationship between OL and organisational trust, continuous improvement, and OP. Recent studies, such as [35,95], confirmed a positive and statistically significant relationship between OL and OP. Meanwhile, some studies reported no relationship between OP and OL [97], while others reported a tenuous relationship between the two variables [93,97,98]. According to the literature, previous studies have concentrated on the contribution of OL to conventional organisational performance, with an absence of research on the possible contribution of OL to organisational sustainability, particularly in an NGO. Given these findings, the current study hypothesised a direct link between OL and an NGO's long-term viability.

**Hypothesis 1 (H1).** *There is a direct interrelationship between OL and SOP.*

### 2.6.2. Organisational Learning, Strategic Human Resource Management, Artificial Intelligence, and Sustainable Organisational Performance

According to [3] learning is viewed as "the next source of sustainable competitive advantage" or "the sole source of competitive advantage" and as the foundation for an organisation's future success [4]. Meanwhile, Ref. [99] suggested that OL implementation could be successful if internal managerial approaches were utilised. This internal management mechanism includes a systematic training process, establishing firm procedures, and a non-rigid work design that entails using a holistic approach to develop human resource strategies that are vertically linked to the company plan. SHRM is a human resource management transformation strategy that focuses on the efficiency of human resource management systems within a business, emphasising human resource collaboration to boost competitive advantage [100,101]. Effective implementation of this strategy requires some input, such as organisational learning, particularly in today's world of increasing competition. Ref. [4] assert that promoting OL demonstrates an effort to create and manage knowledge assets. This demonstrates that OL has the potential to facilitate effective strategic human

resource management. Although studies have established the effect of OL on OP [20,23], Ref. [24] asserts that the mechanism by which OL promotes OP requires further exploration. As a result, it becomes critical to investigate possible mediating variables between OL and OP in response to [21]) contention that their relationship is indirect. Given the growing emphasis on internal operations to enhance the outcomes of firm-level resources and capabilities such as OL, we propose that SHRM, a popular human resources management strategy approach, be included because it integrates with OL and has a synergetic effect on the organisation's long-term performance. Additionally, artificial intelligence is argued to be a mediating variable in this study due to its potential to indispensable factors in firms' development processes, optimisation, and operational flexibility [102]. According to [103], AI increases data processing speed, reducing bottlenecks, and increasing overall operational efficiency by decreasing the time required to process data.

The theoretical understanding of OL's influence on OP appears to be imprecise. However, empirical evidence indicates that OL has a significant effect on OP. Meanwhile, the findings in the literature are inconsistent, particularly regarding the strength of the relationship and the OL and a few selected performance indicators. For example, Refs. [104,105] demonstrated some mediating factors in the relationship between OL and OP. Additionally, Refs. [106,107] demonstrated that OL indirectly affects business performance. This indicates that certain variables may influence the strength of the relationship; thus, we hypothesised the mediating effect of SHRM and AI and developed the following hypotheses:

**Hypothesis 2 (H2).** *OL is directly related to SHRM.*

**Hypothesis 3 (H3).** *There is a direct relationship between OL and AI.*

**Hypothesis 4 (H4).** *The relationship between OL and SOP is partially mediated by (a) SHRM and (b) artificial intelligence.*

2.6.3. Artificial Intelligence and Sustainable Organisational Performance

Furthermore, Refs. [78,108] observed that in today's competitive business environment, every organisation requires the competence of its employees in developing, managing, and implementing intelligent technology that will assist the firm's technical processes in pursuing green initiatives. Similarly, Ref. [109] opined that a firm's growth is contingent upon its ability to acquire and implement AI knowledge in today's contemporary business world. Similarly, Refs. [110,111] observed that numerous task processes can be conceptualised, promoted, and supplemented digitally by collaborative teams working to advance the firm's goals and environmental sustainability objectives. On the other hand, Ref. [108] advocates for a continuous disregard for the digital relevance of any firm's nature of work. As a result, such a business may eventually become irrelevant.

Meanwhile, the literature demonstrates that digitisation has become a global phenomenon promoting shifting labour needs [111]. Notably, with the assistance of AI, several traditionally difficult jobs can be advanced digitally, and [112] observed that the primary digital tasks handled by unique interdependent employees can be nurtured more effectively and efficiently. Furthermore, according to studies, with the assistance of AI, employees' interconnectedness on digital tasks may influence job outcomes and increase productivity [113,114]. In light of this, we propose the following hypothesis:

**Hypothesis 5 (H5).** *There is a direct interrelationship between AI and SOP.*

2.6.4. Strategic Human Resource Management and Sustainable Organisational Performance

SHRM practises are defined as those decisions and actions on the management of employees across all levels of the organisation and those pertaining to implementing strategies aimed at achieving a sustainable competitive advantage [67,115]). This has resulted in an increased focus on the concept by researchers to better understand both the

antecedents and outcomes. Researchers have attempted to explain how SHRM affects form performance. For example, Ref. [116] established a positive and significant relationship between SHRM practises and financial and operational performance of an organisation. Ref. [117] discovered that training and compensation practises result in improved employee performance, resulting in improved organisational performance. Ref. [118] conducted a study in Kenya that established a positive and significant relationship between human resource management and organisational performance. This finding is consistent with several other studies, which have established similar findings in their studies [119,120].

Meanwhile, these studies have been primarily focused on for-profit organisations, leaving scant research on non-profit organisations. Nonetheless, a recent study by [67] examined the impact of SHRM practises on the perceived financial performance of NGOs in Bangladesh and discovered that SHRM practises have a positive effect on perceived financial performance. Given the preceding literature and the fact that NGOs are not profit-driven, it is reasonable to assume that their strategic management must be designed so that the organisation's goals are achieved. As a result, we hypothesised a direct interrelationship between SHRM and SOP.

**Hypothesis 6 (H6).** *There is a direct interrelationship between SHRM and SOP.*

2.6.5. Corporate Social Responsibility, Strategic Human Resource Management, Artificial Intelligence, and Sustainable Organisational Performance

Additionally, CSR is identified as affecting organisational performance [43], owing to its theoretical and practical significance. CSR is generally defined as organisational practises or activities that ensure the fulfilment of a firm's multiple obligations to various stakeholders, including shareholders, employees, customers, the environment, and the local community [44,45]. CSR has become a requirement in recent years due to the goodwill generated by CSR and the interconnectedness of corporate firms and the environment in which they operate. As a result, several attempts have been made to explain the relationship between CSR and OP in terms of achieving competitive advantage [34,57,87,121–125]. For example, Ref. [124] examined the impact of CSR on firm performance and discovered a significant effect, which [125] corroborate.

Similarly, Ref. [121] examined the effect of CSR on OP and established that a firm's public image contributes to CSR, which increases profitability and performance. This position was bolstered by [122], who examined the impact of CSR on the profitability of Nigerian banks and discovered a significant correlation between bank profitability and CSR expenditure. Furthermore, Ref. [121] examined the relationship between CSR and Nigerian firm performance and discovered a positive and significant correlation between the two variables, while [34]) examined the relationship between the two variables in the Jordanian telecommunications sector and discovered a positive effect of CSR on the non-financial and financial performance of telecommunication firms in Jordan. This conclusion was confirmed by [57], who conducted a similar study in the Iranian petroleum industry, and Singh (2021), who examined the Malaysian public sector. Meanwhile, Ref. [87] discovered a strong correlation between CSR, corporate strategy, and competitive advantage in their study.

Additionally, some studies demonstrated the potential benefits of CSR initiatives on organisational outcomes such as commitment, motivation, employee morale, recruitment, loyalty, and turnover, as well as attractiveness to current and prospective employees [43,53,126,127]. According to the reviewed literature, numerous attempts have been made to explain the relationship between CSR and OP. However, in today's new normal, when competition is unpredictable, this is no longer true, and CSR has the potential to influence an organisation's growth, visibility, and sustainability.

Meanwhile, the mechanism by which CSR can ensure the long-term effectiveness of organisational performance has not been thoroughly investigated. As a result, we argue for the inclusion of SHRM and AI in discussing the relationship between CSR and long-term organisational performance. As a result, we propose the following hypotheses:

**Hypothesis 7 (H7).** *There is a direct interrelationship between CSR and SOP.*

**Hypothesis 8 (H8).** *CSR is directly related to SHRM.*

**Hypothesis 9 (H9).** *CSR is directly related to artificial intelligence.*

**Hypothesis 10 (H10).** *The relationship between CSR and SOP is partially mediated by (a) SHRM and (b) artificial intelligence.*

### 3. Research Methodology

As illustrated in Figure 1, this study's conceptual research model demonstrates the relationship between organisational learning (OL), corporate social responsibility (CSR), strategic human resource management (SHRM), artificial intelligence (AI), and sustained organisational performance (SOP). The model establishes a direct relationship between OL and SOP, SHRM, and AI; it also evaluates the direct relationship between AI and SOP, SHRM, and SOP; and it concludes by examining the impact of CSR on SOP, SHRM, and AI. Additionally, this study hypothesised that SHRM and AI plays a moderating role in the relationship between OL and SOP on the one hand, and the relationship between CSR and SOP on the other.

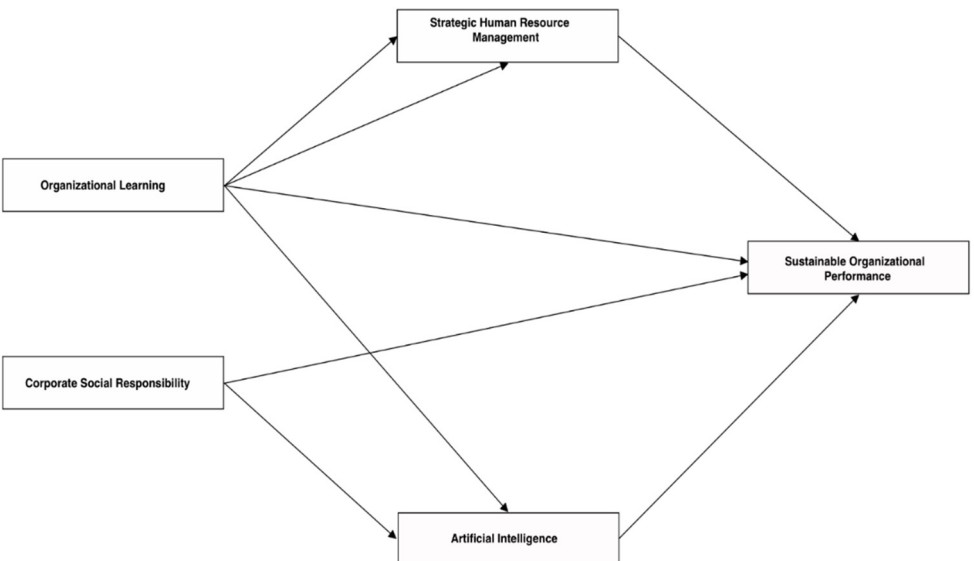

**Figure 1.** Conceptual research model.

### 3.1. Items Measurement

The questionnaire consists of five constructs, each measured by 59 items and graded on a five-point Likert scale ranging from strongly disagree to strongly agree. The materials for this study were obtained through previous research. A total of 6 questions adapted and modified from [25] were used to assess strategic human resource management, 6 items were used to assess artificial intelligence [15], 16 items were used to assess organisational learning [40], 20 items were used to assess CSR [28], and 11 items were used to assess organisational performance [128].

*3.2. Data Collection and Sample Size*

The study's sample included human resource professionals from Jordan's 21 United Nations agencies and 70 non-governmental organisations (NGOs). Thus, a sample size of five human resources employees from each of the 21 UN organisations in Jordan, 105 responses, and 70 from non-governmental organisations was determined. Meanwhile, due to the global pandemic currently ravaging the world, many organisations could not distribute questionnaires in person. Instead, an online survey was used to distribute the questionnaires. Before distributing the questionnaire, an official letter was sent to the heads of all UN organisations and non-governmental organisations in Jordan outlining the study's purpose and requesting permission to collect data.

Additionally, the current investigation adheres to all widely accepted and recognised ethical and approved research protocols. To begin, they ensured the participants' confidentiality and informed them that participation in the study was voluntary but encouraged them to do so. They were then sent a link to the questionnaire, which they were instructed to distribute to the target employees.

## 4. Results and Discussions

SPSS was used to prepare and analyse the data for this study in order to determine the significance of the proposed connections between constructs. The statistical tool "WarpPLS 7.0" was chosen to analyse the structure of this research model in light of [129] research, which stated that the statistical tool is "effective for concurrently analysing non-linear and linear associations". Additionally, PLS-SEM is ideal for ensuring efficacy in assessing the relationship between constructs and producing findings that accurately reflect real-world complexity. Additionally, it is effective when the sample size is small, as this study's sample size was. Finally, the approach takes the non-normality of the data distribution into account.

*4.1. Model Measurements Assessment*

Table 2 summarises the results of the model measure evaluation. According to Table 1, the loadings for items about organisational learning (OL), strategic human resource management (SHRM), artificial intelligence (AI), corporate social responsibility (CSR), and standard operating procedures (SOP) were all greater than the 0.5 thresholds, except for a few items with loadings less than 0.40 that were excluded from further estimation. Additionally, at the 0.001 level of significance, the *p* values associated with these loadings are significant. This finding is consistent with numerous studies [130,131]) demonstrating a high degree of convergent validity for the instrument used to measure the constructs. Additionally, both the Cronbach alpha and composite reliability values for SHRM practises (0.918 and 0.934), AI (0.868 and 0.899), OL (0.890 and 0.913), CSR (0.907 and 0.924), and SOP (0.867 and 0.898) were greater than the suggested threshold value of 0.7, indicating that the instrument of measurement is reliable (Kock, 2014). Additionally, the average variance extracted values for SHRM practises (0.670), AI (0.562), OL (0.572), CSR (0.555), and SOP (0.562) are all greater than the recommended threshold value of 0.5 [130,131], indicating that the measurement has a satisfactory level of internal consistency. Finally, the FVIF values for SHRM (1.926), AI (1.865), OL (2.344), CSR (2.391), and SOP (2.810) are all less than the suggested threshold of 3.3. According to [132], the FVIF coefficient is a "model-wide assessment of multicollinearity that is estimated by combining the variations in the model's other indicators and allows us to determine whether participants perceive our constructs as conceptually distinct from all other constructs".

**Table 2.** Measurements' properties assessment.

| Constructs | Measurement Items | Factor Loading ($\lambda$) | FVIF |
|---|---|---|---|
| Strategic Human Resource Management | ($\propto$ = 0.918; CR = 0.934; AVE = 0.670) | | 1.926 |
| | SHRM1 | 0.812 | |
| | SHRM2 | 0.863 | |
| | SHRM3 | 0.822 | |
| | SHRM4 | 0.843 | |
| | SHRM5 | 0.831 | |
| | SHRM6 | 0.775 | |
| Artificial Intelligence | ($\propto$ = 0.868; CR = 0.899; AVE = 0.562) | | 1.865 |
| | AI1 | 0.622 | |
| | AI2 | 0.780 | |
| | AI3 | 0.809 | |
| | AI4 | 0.801 | |
| | AI5 | 0.758 | |
| | AI6 | 0.792 | |
| Organisational Learning | ($\propto$ = 0.890; CR = 0.913; AVE = 0.572) | | |
| | OL1 | 0.713 | |
| | OL2 | 0.734 | |
| | OL3 | 0.736 | |
| | OL4 | 0.754 | |
| | OL5 | 0.775 | |
| | OL6 | 0.551 | |
| | OL7 | 0.839 | |
| | OL8 | 0.767 | |
| | OL9 | 0.324 * | |
| | OL10 | 0.746 | |
| | OL11 | 0.521 | |
| | OL12 | 0.689 | |
| | OL13 | 0.683 | |
| | OL14 | 0.201 * | |
| | OL15 | 0.851 | |
| | OL16 | 0.891 | |
| Corporate Social Responsibility | ($\propto$ = 0.907; CR = 0.924; AVE = 0.555) | | 2.391 |
| | CSR1 | 0.825 | |
| | CSR2 | 0.639 | |
| | CSR3 | 0.731 | |

**Table 2.** *Cont*.

| Constructs | Measurement Items | Factor Loading ($\lambda$) | FVIF |
|---|---|---|---|
| | CSR4 | 0.689 | |
| | CSR5 | 0.575 | |
| | CSR6 | 0.700 | |
| | CSR7 | 0.767 | |
| | CSR8 | 0.727 | |
| | CSR9 | 0.833 | |
| | CSR10 | 0.848 | |
| | CSR11 | 0.762 | |
| | CSR12 | 0.322 * | |
| | CSR13 | 0.648 | |
| | CSR14 | 0.657 | |
| | CSR15 | 0.789 | |
| | CSR16 | 0.890 | |
| | CSR17 | 0.910 | |
| | CSR18 | 0.830 | |
| | CSR19 | 0.910 | |
| | CSR20 | 0.907 | |
| Sustainable Organisational Performance | ($\propto$ = 0.867; CR = 0.898; AVE = 0.562) | | 2.810 |
| | SOP1 | 0.467 | |
| | SOP2 | 0.790 | |
| | SOP3 | 0.889 | |
| | SOP4 | 0.984 | |
| | SOP5 | 0.677 | |
| | SOP6 | 0.890 | |
| | SOP7 | 0.763 | |
| | SOP8 | 0.567 | |
| | SOP9 | 0.801 | |
| | SOP10 | 0.911 | |

Note: (1) SHRM = strategic human resource management; AI = artificial intelligence; OL= organisational learning; CSR = corporate social responsibility; SOP = sustainable organisational performance. (2) AVE = average variance extracted; CR = composite reliability; $\alpha$ = Cronbach alpha. (3) * denotes the items removed.

After examining the measurement instrument's reliability, we examined the variables' discriminant validity. According to Table 3, the "square root of the average variance extracted and presented in diagonal" for each construct must be greater than the correlations between that construct and the other constructs [133]. Our findings indicate that SHRM practice, AI, OL, CSR, and SOP all have a high level of discriminant validity in our model.

**Table 3.** Correlations among 1.vs with sq. rts. of AVEs.

|  | SHRM | AI | OL | CSR | SOP |
|---|---|---|---|---|---|
| SHRM | **0.819** | | | | |
| AI | 0.645 | **0.749** | | | |
| OL | 0.103 | 0.084 | **0.756** | | |
| CSR | 0.115 | 0.113 | 0.672 | **0.745** | |
| SOP | 0.220 | 0.162 | 0.707 | 0.718 | **0.749** |

Note: SHRM = strategic human resources management, AI = artificial intelligence, OL = organisational learning, CSR = corporate social responsibility, SOP = sustainable organisational performance. Square roots of average variances extracted (AVEs) illustrated on diagonal.

### 4.2. Common Bias Method (CMB)

According to Kock (2015), the coefficients of complete collinearity VIF are highly susceptible to "pathological common variations" in all variables in methodological settings, which is consistent with the findings in this work. Therefore, even if a model passes discriminant and convergent validity tests, it is necessary to examine the CMB due to its susceptibility to common pathological variations. Given this, the literature indicates that a threshold value of 5 is acceptable and that a value of 3.3 is optimal for the full collinearity VIF value [131,132]). As a result, when the entire set of VIF values shown in Table 1 is used, all contrasts exceed the permissible threshold.

### 4.3. Testing of Hypotheses

We examined the model fit indices summarised and presented in Table 4 prior to interpreting the path coefficients of our model's proposed relationships. When the indices are significant or satisfy the applicable criterion, the structural model is of sufficient quality [134,135]). Table 4 demonstrates that the structural model used in this study is both fit and adequate in its entirety.

**Table 4.** Structural model fit.

| Indices | Coefficient | Decision |
|---|---|---|
| APC | 0.182 | $p < 0.001$ |
| ARS | 0.579 | $p < 0.001$ |
| AVIF | 2.398 | Acceptable if $\leq 5$, ideally $\leq 3.3$ |
| AFVIF | 3.293 | Acceptable if $\leq 5$, ideally $\leq 3.3$ |
| GOF | 0.667 | Small $\geq 0.1$, medium $\geq 0.25$, large $\geq 0.36$ |
| RSCR | 0.919 | Acceptable if $\geq 0.9$, ideally =1 |
| SRMR | 0.096 | Acceptable if $\leq 0.1$ |
| SChS | 1.233 | $p < 0.001$ |

Following validation of our model's fitness, we examined the significance of our constructs' linear and non-linear interconnections. As shown in Figure 2, OL and CSR account for approximately 2% of the explained variation in SHRM, accounting for approximately 4% of the explained variation in artificial intelligence. OL, CSR, SHRM, and AI account for approximately 9% of the variation in the SOP's explanations. These $R^2$ values, according to [136], indicate that the formative constructs account for a small proportion of the variance. As shown in Table 5, the model testing results indicate that both the coefficient path and *p*-value for H1 (= 0.16, $p = 0.02$) and H3 (= 0.17, $p = 0.01$) are positive and significant. As a result of accepting H1 and H3, we conclude that organisational learning has a direct and positive relationship with long-term organisational performance and artificial intelligence that is statistically significant at the less than 5% level. Meanwhile, as shown in Table 5,

the coefficient and *p*-value for H2 (=0.11, *p* = 0.08) indicate that the *p*-value exceeds the 5% confidence level, and thus the hypothesis is rejected. Additionally, the *p* values for the H4a and H4b coefficients indicate that H4a (=0.24, *p* = 0.01) and H4b (=0.56, *p* = 0.03) are statistically significant.

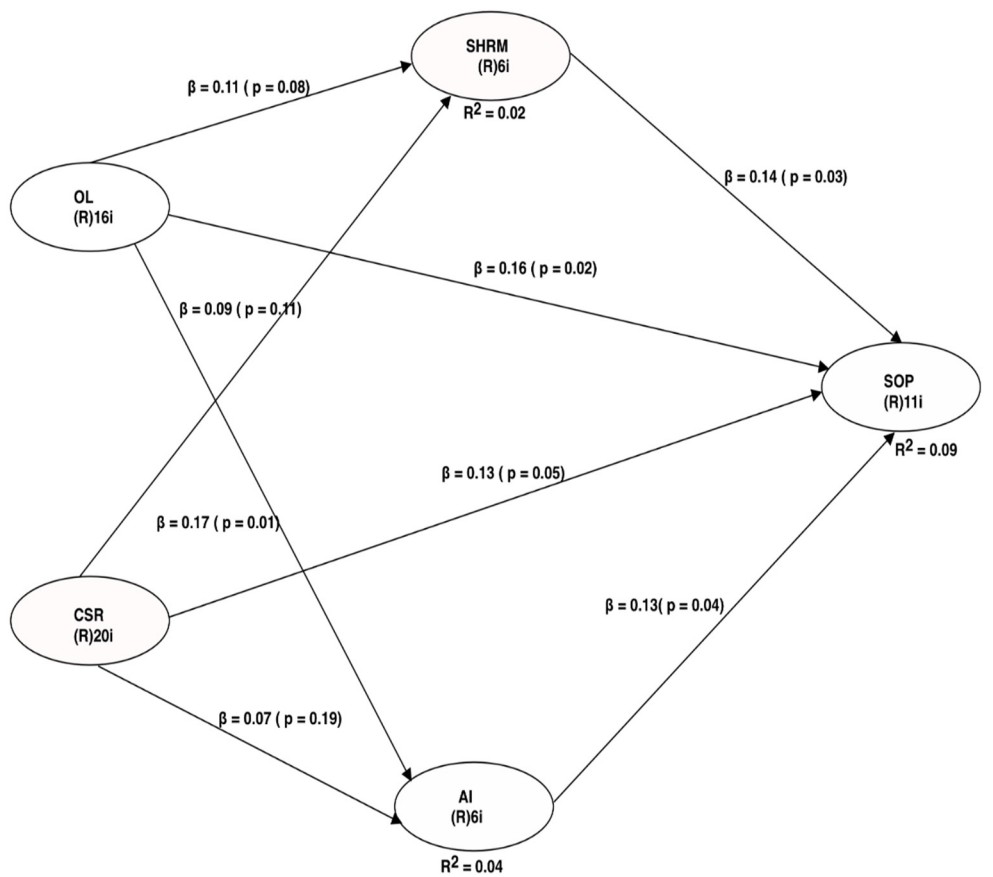

**Figure 2.** Model Testing results.

**Table 5.** Coefficients and *p* values Path.

| Hypothesis | Interaction | Path Coefficient | *p* Value | Decision |
|:---:|:---:|:---:|:---:|:---:|
| H1 | OL → SOP | 0.16 | 0.02 | Accepted |
| H2 | OL → SHRM | 0.11 | 0.08 | Rejected |
| H3 | OL → AI | 0.17 | <0.01 | Accepted |
| H4a | OL → SHRM → SOP | 0.24 | <0.01 | Accepted |
| H4b | OL → AI → SOP | 0.56 | 0.03 | Accepted |
| H5 | AI → SOP | 0.13 | 0.04 | Accepted |
| H6 | SHRM → SOP | 0.14 | 0.03 | Accepted |
| H7 | CSR → SOP | 0.13 | 0.05 | Accepted |
| H8 | CSR → SHRM | 0.09 | 0.11 | Rejected |
| H9 | CSR → AI | 0.07 | 0.19 | Rejected |
| H10a | CSR → SHRM → SOP | 0.26 | <0.01 | Accepted |
| H10b | CSR → AI → SOP | 0.17 | <0.01 | Accepted |

Note: SHRM = strategic human resources management, AI = artificial intelligence, OL = organisational learning, CSR = corporate social responsibility, SOP = sustainable organisational performance.

As a result, we accept H4a and H4b and conclude that strategic human resource management and artificial intelligence are mediators between OL and SOP. This finding demonstrates that OL affects SOP both indirectly and directly via strategic human resource practises and artificial intelligence.

In addition, results of the influence of artificial intelligence, SHRM, and CSR on SOP as hypothesised in H5 ($\beta = 0.13$, $p = 0.04$), H6 ($\beta = 0.14$, $p = 0.03$), and H7 ($\beta = 0.13$, $p = 0.05$), respectively, shows that they all have a significant influence on SOP. Thus, we accept H5, H6, and H7 and conclude that artificial intelligence, strategic human resources management practices, and corporate social responsibility significantly influence sustainable organisational performance with less than 5% significant level. However, the direct relationship between CSR and SHRM (H8: $\beta = 0.09$, $p = 0.11$) on the one hand, and CSR and AI (H9: $\beta = 0.07$, $p = 0.19$) on the other hand were found not to be statistically significant; hence we reject H8 and H9. However, the mediating role of SHRM and AI in the interrelationship between CSR and SOP has been hypothesised in H10a ($\beta = 0.26$, $p < 0.01$) and H10b ($\beta = 0.17$, $p < 0.01$) were found to be statistically significant. As a result, we accept H10a and H10b and conclude that SHRM and AI serve as a conduit for the interaction of CSR and SOP. This implies that CSR affects SOP both indirectly and directly.

## 5. Discussions and Conclusions

Significant emerging concepts, OL and CSR, were examined within the context of SOP to position them in terms of ecological quality while achieving a sustainable competitive edge through SHRM practises and AI. This was done to investigate OL and CSR as predictors of long-term organisational performance and to shed light on how SHRM practises and AI can aid in SOP implementation. This study demonstrates that OL and CSR have a positive and significant impact on Jordan's standard operating procedures. The positive correlation between OL and SOP observed in this study is consistent with previous research [90–92,96]. On the other hand, this study examined the effect of OL on organisational performance and sampled only profit-oriented organisations.

Meanwhile, the results contradict the findings of [97,99] who concluded in their studies that no relationship exists between the two variables. According to the literature, organisational learning is the "ability of an organisation to process knowledge—that is, to create, acquire, transmit, and integrate information—and to adapt its behaviour to the current cognitive environment to maximise performance". Our findings support the notion that organisations can achieve sustainable performance by implementing OL effectively. Similarly, the present study's finding of a significant and positive relationship between CSR and SOP is consistent with findings from a number of prior studies on profit-driven organisations [34,57,123–125] This demonstrates that an NGO's performance can be sustained by adhering to ethical behaviour aimed at economic advancement while also promoting employees' quality of life, their families, the local community, and society in general.

Considering the literature's evidence of SHRM's contribution to organisational performance, this study demonstrates how SHRM practises can assist organisations in maintaining their performance. This finding is consistent with several previous studies [9,116,118–120,137], though in for-profit organisations, as well as with [67], who conducted a similar study using Bangladeshi NGOs. Additionally, we discovered that artificial intelligence is critical for sustaining organisational performance over the long term. This finding is consistent with [112,138,139], who posited that allowing employees to perform certain tasks digitally enables more effective communication and knowledge exchange, which is capable of promoting innovative work behaviours that result in the firm gaining an advantage over the competition and thus contributing to the sustainability of their performance.

Furthermore, it was discovered that the moderating effects of SHRM and AI were beneficial in enhancing the positive influence of OL on SOP on the one hand and the relationship between corporate social responsibility and SOP on the other. This finding cor-

roborates the findings of [104–106,108], who all suggested that a variety of factors mediate the relationship between OL and SOP in their respective studies. Effectively implementing SHRM, which defines the structure of the plan by which the firm's human resources intend to accomplish organisational goals, will ensure the organisation's performance is sustainable. Additionally, as demonstrated in this study, AI can amplify the effect of OL on SOP by enabling effective communication and knowledge exchange between employees [112,138].

While numerous studies have examined both the outcomes and antecedents of generic organisational performance, our research is the first to examine the antecedents of SOP in non-governmental organisations and the moderating effects of SHRM practises and artificial intelligence. This study sheds new light on how OL and CSR can assist Jordan's non-governmental organisations achieve sustainable performance. This study was conducted in a developing economic context (Jordan), with a special emphasis on NGOs, which have been underrepresented in previous studies to generate substantial and original theory and practice knowledge. While prior research on OL and CSR has yielded important findings, none has examined how OL and CSR contribute to the sustainability of NGO performance. This study aimed to fill a void in the literature by situating OL and CSR within the context of contemporary environmental sustainability. Additionally, the empirical findings that SHRM practises and AI are effective for organisational performance were strengthened by demonstrating in this study that these variables amplify the positive effect of OL and CSR on organisational sustainability. Significant insights have been gleaned from rapidly developing areas of the literature due to our discoveries, allowing us to build on the scholars' innovative discoveries and contribute to contemporary understanding of OL, CSR, SHRM, AI, and sustainable development organisational performance in the context of NGOs.

Our study's findings also have significant implications for practitioners and industry policymakers in developing guidelines to ensure the effective implementation of OL and CSR within their respective organisations to achieve the organisation's sustainability goals. To ensure the long-term viability of an organisation's performance, such as an NGO, it is necessary to establish an effective mechanism for ensuring the development of their knowledge assets and putting forward practical methods for managing them. Additionally, they should ensure that they are fulfilling their numerous responsibilities to various stakeholders, including employees, shareholders, customers, the environment, and local communities. This becomes critical when one considers the organisational structure of NGOs, the primary mission of which is to volunteer in civil society in response to state failures. To accomplish this goal, NGOs must manage their resources efficiently to ensure their sustainability [28].

Additionally, NGOs should ensure that SHRM is implemented effectively to maximise the impact of organisational learning and corporate social responsibility on their performance sustainability. Similarly, the evidence for artificial intelligence's importance as a predictor and moderator of long-term organisational performance demonstrates the importance of AI in an organisation. Finally, an organisation's strategy must be crystal clear to achieve its objectives, which can be accomplished only when its managers mobilise the necessary human, technological, and financial resources. The majority of challenges can be addressed by integrating AI into an organisation by optimising processes and organisational performance. As a result, integrating AI into an organisation necessitates training employees to ensure the quality of future jobs in a world where humans and machines coexist. To maintain confidence, businesses should establish an "external and internal control tower" for data ethics, recruit and retain technologically savvy employees, and adapt training tools to accommodate increasing training volume and variety.

*Limitations and Future Research*

Although several experts have argued in favour of OL as a counterbalance to the myriad challenges confronting organisations, managers have yet to completely grasp the concept. However, continuous learning is necessary for organisational improvement, and a lack of understanding has made it difficult to spread and implement OL and other critical factors affecting organisational performance sustainability. This has posed a significant challenge for businesses, particularly non-profit organisations, which pursue growth and development in a manner distinct from profit-oriented organisations to maintain a competitive edge in dynamic environmental conditions. Considering these factors, this study established that OL, CSR, SHRM practises, and AI are all significant predictors of long-term organisational performance. Additionally, this study discovered that SHRM practises and artificial intelligence can potentially mitigate the effect of OL and CSR on long-term organisational performance. By incorporating additional relevant indicators, the model developed in this study paves the way for researchers and practitioners to develop more complex, holistic, and comprehensive models to investigate CRS and OL's outcomes or other SOP antecedents. Meanwhile, the study's limitations include using a non-probability sampling technique and a narrow focus on a single industry, limiting the findings' generalisability. As a result, replicating the model in various industries will be fascinating.

**Author Contributions:** Conceptualization, A.A. and H.R.; methodology, A.A.; software, A.A.; validation, A.A, H.R. and H.O.; formal analysis, A.A.; investigation, A.A.; resources, A.A.; data curation, A.A.; writing—original draft preparation, A.A.; writing—review and editing, H.R. and H.O.; visualization, H.R. and H.O.; supervision, H.R. and H.O.; and project administration, H.R., H.O. All authors have read and agreed to the published version of the manuscript.

**Funding:** This research received no external funding.

**Institutional Review Board Statement:** Not applicable.

**Informed Consent Statement:** Not applicable.

**Data Availability Statement:** Data is readily available at the request from the first author.

**Conflicts of Interest:** The authors declare no conflict of interest.

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
