# Peer review of "Do Strategic Human Resources and Artificial Intelligence Help to Make Organisations More Sustainable? Evidence from Non-Governmental Organisations"

_sustainability, doi:10.3390/su14127327_

Round 1

Reviewer 1 Report

I recommend to the authors the clear delimitation of the research hypotheses from the literature review. For this, I propose to include the hypotheses in the Research Methodology section.

Restructuring Table 1 so that there are no more blank columns.

In the manuscript file Figure 1 was not visible and its positioning is before the beginning of the section in which it is mentioned.

Author Response

Dear Professor

Thank you very much for your invaluable comments and suggestions, which have improved the revised version significantly.

We would also like to send our appreciation to you for your time and efforts in reviewing our paper and for providing the excellent comment. 

We hope that the revised version of the manuscript meets your expected comments and suggestions.

Kind Regards

Reviewer 2 Report

This paper is very interesting and novel, but   the authors might consider the following question before publication

-A table that includes similar studies carried out to date could improve the theoretical framework

-Authors could add scales with items for each of the variables. 

-The figure looks split

-Justify each hypothesis separately. This would provide a better understanding of each of them

Author Response

(The authors gave the same response as above.)

Reviewer 3 Report

The article addresses an interesting aspect related to the use of strategic human resource management and artificial intelligence to increase the sustainable development of the organization.

The presented literature review is appropriate from the point of view of the purpose of the article. It was based on a very large, fully satisfactory number of sources, which confirms the authors' in-depth analysis of the problem.

Development of Hypotheses is generally correct. The hypotheses posed are well supported by the analysis of the literature.

The results are generally presented in a legible manner and the conclusions drawn are correct and fairly well supported by the results of research of other authors.

In the file available to me, picture 1 was mostly invisible, so it is difficult for me to relate to its content.

Author Response

(The authors gave the same response as above.)

Round 2

Reviewer 2 Report

The authors have made the suggestions made.